# Information Systems Strategy for Multi-National Corporations: Towards an Operational Model and Action List

Martin Wynn [1],* and Christian Weber [2]

1   The School of Business, Computing and Social Sciences, University of Gloucestershire, Cheltenham GL50 2RH, UK
2   Independent Researcher, 8582 Dozwil, Switzerland; christian.weber99@outlook.com
*   Correspondence: mwynn@glos.ac.uk

**Abstract:** The development and implementation of information systems strategy in multi-national corporations (MNCs) faces particular challenges—cultural differences and variations in work values and practices across different countries, numerous technology landscapes and legacy issues, language and accounting particularities, and differing business models. This article builds upon the existing literature and in-depth interviews with eighteen industry practitioners employed in six MNCs to construct an operational model to address these challenges. The research design is based on an inductive, qualitative approach that develops an initial conceptual framework—derived from the literature—into an operational model, which is then applied and refined in a case study company. The final model consists of change components and process phases. Six change components are identified that drive and underpin IS strategy—business strategy, systems projects, technology infrastructure, process change, skills and competencies, and costs and benefits. Five core process phases are recognized—review, align, engage, execute, and control. The model is based on the interaction between these two dimensions—change components and process phases—and an action list is also developed to support the application of the model, which contributes to the theory and practice of information systems deployment in MNCs.

**Keywords:** information systems strategy; multi-national corporation; MNCs; conceptual framework; operational model; action list

## 1. Introduction

Strategic alignment in multi-national corporations (MNCs) has been the subject of academic and industry-based research for several decades [1,2], and in recent years, this has included a focus on information systems [3]. Chen et al. [4] (p. 233) define information systems (IS) strategy as "an organizational perspective on the investment in, deployment, use, and management of IS". The main objective is the provision and operation of IS that support evolving business requirements and align with the overall business strategy. This alignment is particularly complex when a company operates in an international context and has a multi-subsidiary business model. Where such companies have accomplished a high degree of alignment, more effective system deployment generally results in improved business efficiencies and performance [5]. One major issue is the range of cultures that an MNC often has to accommodate in implementing an IS strategy. Adaba et al. [6] (p. 288), for example, in their study of the impact of national culture on strategic IT alignment in MNCs, found that "national cultures affect alignment indirectly, through variables in the organizational context, including intercultural communications, IT governance, management style, differences in work values and practices, and cultural conflict and mistrust". A further challenge is managing the growing complexity and range of digital technologies that are being introduced alongside or within the main corporate business systems. The use of advanced analytics, artificial intelligence applications, and the data capture capabilities of Internet of Things (IoT) devices, for example, have implications for IS selection and operation, as

do the various options for IS operation in the cloud [7]. This includes the threat posed by cybersecurity failure. In their study of enterprise information systems, for example, Singh et al. [8] (p. 1) note that "as organizations communicate more, networks have become more open within enterprises' facilities and their vendors, dealers, and customers. Eventually, enterprises become exposed and increasingly susceptible to information leakages, data thefts, cyber-attacks, and sabotage".

Although there are a number of models in the extant literature that focus on either IS strategy development or IS strategy implementation, very few look at both in harness, particularly in the context of MNCs. Few of the existing IS strategy frameworks include any implementation considerations, such as timing, detailed actions, deliverables, specific roles, and responsibilities. Peppard et al. [9] observed that very little work had been conducted focusing on IS strategy as a micro or social process, and there remains little guidance on when to execute a step or a specific phase in strategy implementation, with approaches tending to be generic, with no specification of required deliverable(s). The literature search failed to find any integrated model or framework for IS development and implementation in MNCs, and this research aims to fill this gap in the literature and in practice by providing a framework that encompasses and combines IS strategy development and implementation.

The article addresses the following research questions (RQs):

RQ1. From an analysis of the extant literature, what conceptual framework can be constructed to guide IS strategy development and implementation in MNCs?

RQ2. What does feedback from practitioner interviews indicate regarding the relevance and usability of the framework?

RQ3. Can a set of key actions be developed to support the use of the framework in practice?

Following this introduction, the article comprises four further sections. In Section 2, the three phases of the research are outlined, and this is followed by a review of relevant literature and the development of the conceptual framework. Section 4 sets out and discusses the main results from the matching of the interview material to the conceptual framework, assesses the application of the framework in a case study company, and presents an action list to support the use of the framework in practice. Finally, the conclusion section summarizes responses to the research questions, discusses limitations, and points up possible future avenues of research in this field.

## 2. Research Method

This research adopts an interpretivist philosophy and is based on a scoping literature review to construct the initial conceptual framework, in-depth interviews with industry practitioners to provide material to populate the framework, and an applied case study to progress the development of an action list to support the application of the framework (Figure 1). This is essentially qualitative research, which Mason [10] concluded, because of its intensity, provides a powerful source of information for analysis. Each of these three research phases is outlined below.

First, the extant literature was assessed. Bell et al. [11] (p. 97) have observed that a literature review can provide "a means of gaining an initial impression" of relevant themes and that "the narrative review may be more suitable for qualitative or inductive researchers, whose research strategies are based on an interpretative epistemology". This was a scoping literature review aimed at identifying the key themes from the literature that could support the construction of a conceptual framework for the primary research. A scoping review involves a "broad scan of contextual literature" through which "topical relationships, research trends, and complementary capabilities can be discovered" [12] (p. 351). It provides an initial overview of the subject matter "to draw the big picture" [13] (p. 1). Various academic databases, including IEEE Xplore, Google Scholar, Scopus, Web of Science, Science Direct, and JSTOR, were accessed to search the existing literature. This allowed the identification of a set of key issues and related questions that were subsequently developed in the interview questionnaire.

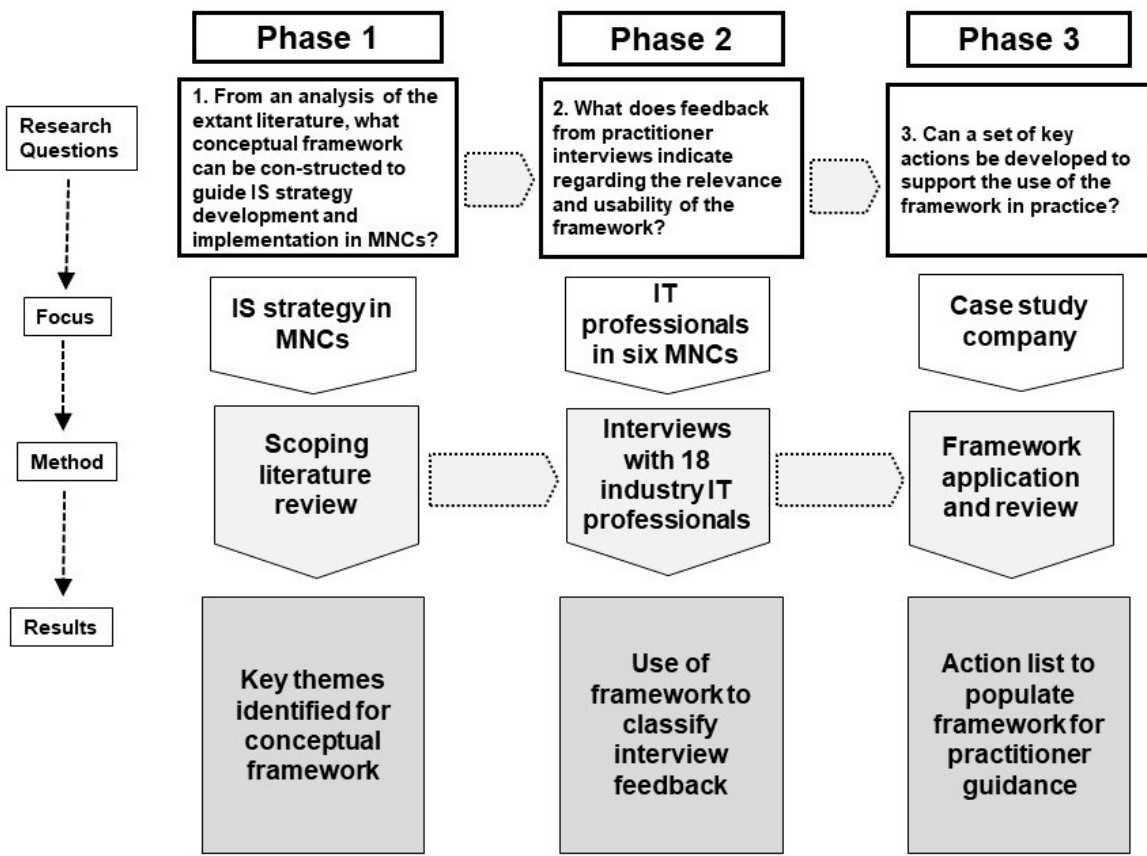

**Figure 1.** Research method: the three phases of research.

Second, interviews were undertaken with expert practitioners. Through the professional networks of the authors, purposive sampling was used to select the 18 interviewees spread across six MNCs (Table 1) who were "chosen because they have particular features or characteristics which will enable detailed exploration and understanding of the central themes and puzzles which the researcher wishes to study" [14] (p. 78). The researchers considered that after 18 interviews, further interviews were unlikely to provide more information. This is in line with Kuzel's [15] study of sampling in qualitative inquiry, which recommended six to eight interviews for a homogenous sample, and 12 to 20 when "trying to achieve maximum variation" (p. 41). Similarly, Guest et al. [16] (p. 78) found that "a sample of six interviews may have been sufficient to enable development of meaningful themes and useful interpretations".

Interviewees were sent a pre-interview questionnaire containing a set of 23 questions linked to the three main research questions and the conceptual framework. Interviews were held face-to-face or via TEAMS/SKYPE and lasted at least one hour each. This was considered the best way of eliciting qualitative data, with the highest possible level of knowledge being acquired in a flexible manner, giving interviewees a "voice" in the study [17]. A combination of middle and senior managers, working in posts related to IS strategy (Table 2), provided a range of perspectives on the main themes developed in the conceptual framework. The companies for which the interviewees worked are anonymized. They had global operations but had their headquarters in Europe. Of the 18 interviewees, 16 were European and 2 were Asian. Cultural differences noted in interviewee responses are based on their experience and perceptions regarding IS strategy formulation and implementation within their company, rather than reflecting their own behavior based on cultural differences. The conceptual framework was used as a reference point for analyzing and classifying key themes identified in the interview material, which provided the basis for assessing its value and relevance.

**Table 1.** Companies featuring in phase 2 of the research.

| Company Code | Industry Sector | Subsidiaries | Employees | Strategy Type |
|---|---|---|---|---|
| A | Industrial machinery | 140 | 13,000 | Global |
| B | Semiconductors | 5 | 1700 | Global |
| C | Multi-sector holdings | 400 | 77,000 | Multi-domestic |
| D | Metal and glass containers | 10 | 800 | Global |
| E | Application software | 77 | 4000 | Transnational |
| F | Electrical components and equipment | 60 | 4000 | Global |

**Table 2.** Interviewees in phase 2 of the research.

| Code | Current Job Role/Position | Years in Management | Years in Company | Years in Current Position |
|---|---|---|---|---|
| P1 | CIO | 20 | 20 | 14 |
| P2 | Global Head Digital Supply Chain Management | 10 | 10 | 2 |
| P3 | General IT Manager | 14 | 24 | 14 |
| P4 | Head of Information Security | 5 | 12 | 1 |
| P5 | Head of Projects and Processes | 15 | 15 | 1 |
| P6 | Group IT Manager | 20 | 20 | 20 |
| P7 | Head of IT Operations | 6 | 9 | 1 |
| P8 | Head of Customer Experience Applications | 5 | 9 | 2 |
| P9 | Head of IT Basis and Client Services | 10 | 15 | 10 |
| P10 | Head of Data Management | 25 | 8 | 5 |
| P11 | Head of Business Processes and Applications | 10 | 20 | 4 |
| P12 | CIO | 15 | 14 | 11 |
| P13 | IT Manager | 2 | 4 | 1 |
| P14 | Head of Treasury Operations | 7 | 5 | 5 |
| P15 | Head of IT Business Development | 2 | 4 | 4 |
| P16 | Operations Manager | 25 | 25 | 20 |
| P17 | CFO | 15 | 4 | 4 |
| P18 | Operations Manager | 13 | 13 | 8 |

The content analysis of the questionnaire responses and the 18 semi-structured expert interviews initially resulted in 270 codes in NVivo (numbered Rxx). Subsequent synthesis of the data, thematic analysis, and data reduction allowed a consolidation of just 62 codes applied to 192 responses that were assigned to the appropriate cell in the conceptual framework matrix. Table 3 provides an example of the codes allocated to the technology infrastructure change component. This allowed identification of the key phases for each change dimension in the framework and also provided some key feedback that was used in developing the action list noted below and detailed in Section 4. Interviews were held in English, recorded, and transcripts were made. NVivo was used along with SPSS to collect and analyze the primary data and document the answers to address the research questions. Neither the organizations' names nor the participants' names are mentioned in this research study.

**Table 3.** Data coding: codes allocated to the technology infrastructure change component.

| Consolidated R Code | Type | Change Component | Process Phase |
|---|---|---|---|
| R62—Knowing why and where to go and how to use tools. | Response | Technology | 1. Review |
| R62—IS is the key to drive future actions. | Response | Technology | 1. Review |
| R62—Simple strategy documentation and presentation. | Response | Technology | 2. Align |
| R62—A good model is the most important. | Response | Technology | 2. Align |
| R61—Information sharing. | Response | Technology | 2. Align |
| R61—Considering digital natives and how they change or affect the IS strategy. | Response | Technology | 2. Align |
| R61—Common agreement of terms and definitions. | Response | Technology | 2. Align |
| R46—See CIO PPTX for 2025. | Response | Technology | 2. Align |
| R46—SAP rollout in more countries. | Response | Technology | 2. Align |
| R46—Rollout more functions in O365 | Response | Technology | 2. Align |
| R46—Optimization of current processes and templates. | Response | Technology | 2. Align |
| R46—More cloud services. | Response | Technology | 2. Align |
| R46—More business com tool like S4b of teams. | Response | Technology | 2. Align |
| R45—See strategy from CIO PPTX. | Response | Technology | 2. Align |
| R45—SAP rollout in more countries. | Response | Technology | 2. Align |
| R44—Too many systems and processes not aligning resulting in island solution; breaking this up takes CEO or EXEC board actions. | Response | Technology | 2. Align |
| R44—Finding right balance implementing new technologies and organization readiness. | Response | Technology | 2. Align |
| R35—Siebel. | Response | Technology | 2. Align |
| R35—SAP. | Response | Technology | 2. Align |
| R35—O365. | Response | Technology | 2. Align |
| R35—Navision dynamics. | Response | Technology | 2. Align |
| R35—Jaggaer. | Response | Technology | 2. Align |
| R35—ISO 27001. | Response | Technology | 2. Align |
| R34—When trends arise it triggers a strategy review or change. | Response | Technology | 2. Align |
| R34—Finding megatrends and try integrate them into the strategy. | Response | Technology | 2. Align |
| R31—Considering technology changes and current architecture drives development. | Response | Technology | 2. Engage |
| R31—Current Doc review to outline key systems and technologies. | Response | Technology | 2. Engage |
| R44—Less training resulting in poor system usage. | Response | Technology | 4. Execute |
| R62—Need for speed. | Response | Technology | 5. Control |
| R44—Taking assumptions and not facts resulting in project restart. | Response | Technology | 5. Control |

Thirdly, the conceptual framework was applied in practice in a case study organization to develop a set of actions relating to each cell in the conceptual framework matrix. This was built upon the feedback and insights gained from the practitioner interviews. The validity and generalizability of case studies have been discussed widely in the literature [18,19]. Flyvbjerg [20] has suggested that cases should not necessarily be used for generalization beyond the case study environment studied, but rather should focus on the generation of a deep understanding of the complexity of the case, producing "concrete, context-dependent knowledge" (p. 223). Here, the applied case study is used to complement material gleaned from the interviews to develop the action list and also provides some validation of the framework in practice, albeit in just one case study.

## 3. Relevant Literature and Conceptual Framework

This section comprises three sub-sections. First, the relevant literature, from which the two-dimensional conceptual framework is developed, is briefly reviewed. Then, in Section 3.2, the change components—one of the two dimensions of the conceptual framework—are outlined and discussed. This is followed in Section 3.3 by a description of the process phases, the second dimension in the conceptual framework.

### 3.1. Relevant Literature

The conceptual framework builds upon an analysis of the relevant literature. Whitten [21] saw IS as an integrated web of people, processes, data, software, hardware, and procedures that interact with each other to analyze and distribute collected and processed information, create value, and support the systems inside and outside an organization. Laudon and Laudon [22] identified three core dimensions—organization, management, and technology—for IS strategy development, allowing managers, project managers, process owners, and employees to use information systems more efficiently. Stair and Reynolds [23] envisaged the successful implementation of an IS strategy as a process of mutual transformation; the organization and the technology transform each other during the implementation process; and Wang et al. [24] (paragraph 8) note that "the overall IS can guarantee the realization of enterprise performance goals in the aspects of organizational design, resource allocation, and management improvement". More specifically, Vaidya [25] suggested there are five main factors involved in technology strategy development for multinational companies: technical considerations, operational considerations, economic considerations, social factors, and the political environment. Vaidya [25] (p. 7) observes that "operational considerations are influenced by social and political factors in a multinational company. They are crucial for a company having manufacturing and/or service centers in multiple countries and customers spread globally". The author put forward a model—the IPCRC model (Figure 2)—for developing technology strategy in MNCs. One aspect the author highlighted was centralization vs. decentralization of systems: "the strategist needs to be conscious of the impact of decisions about centralizing and decentralizing systems. The centralized system gives an advantage of uniformity and cost reduction, but also increases challenges of satisfying requirements of diverse set of people with diverse requirements" [25] (p. 10).

The alignment of strategy from the center to the subsidiary and from business to IT is a central theme evident in the literature. Ali et al. [26] (p. 5) observe that "the extant literature advocates that the alignment between IT governance and organizational capabilities and strategies has a meaningful impact on business outcomes" and that "alignment in this area generates competitive advantages, while misalignment can spawn negative consequences". In the context of business strategy in MNCs, Galliers and Leidner [27] identified three main organizational structures—centralized, decentralized and federated—and suggested there were different strategic, tactical, and control processes for each organization type. More specifically, regarding the IT domain, Earl [28] introduced the differentiation between information technology (IT), information systems (IS), and information management (IM) strategies and suggested there were three ways in which these strategies could be aligned

with overall business strategy—top down, bottom up, and inside out. Subsequent studies have shown that the lack of alignment of these strategies with overall business strategy is one of the main reasons why enterprises fail to exploit the full potential of their investment in information technology and systems [29].

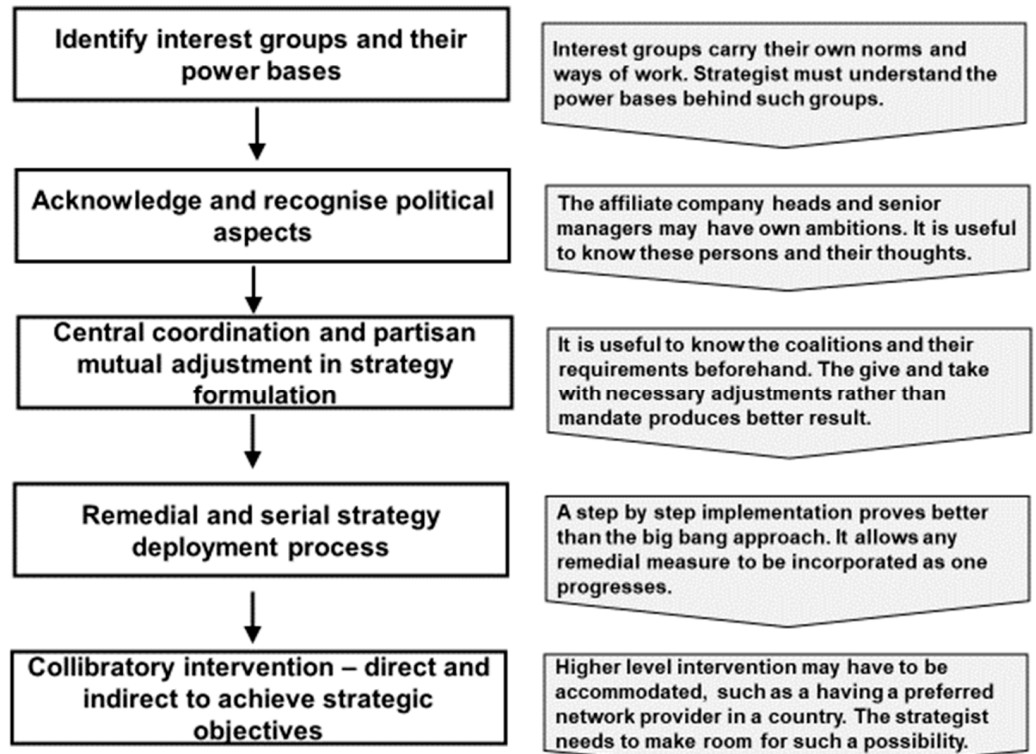

**Figure 2.** Vaidya's IPCRC model for technology strategy development in MNCs. Adapted from Vaidya [25] (p. 12).

A number of approaches have been put forward to tackle the alignment issue. One of the most cited is the strategic alignment model (SAM) of Henderson and Venkatraman [30], which has two main dimensions—strategic fit and functional integration—in a four-quadrant model. De Castro et al. [31] used model-driven architecture as a tool to analyze and enable alignment, and Aversano et al. [32] developed a framework for modeling functional alignment and measuring the degree of alignment between business processes and software systems. More recently, Peppard and Fonstad [33] (paragraph 3) have suggested a new perspective on the alignment challenge: instead of attempting to align IT and business strategies in a formal manner, the authors maintain that "coevolving digital with customers and ecosystem partners" is now more appropriate in the digital era.

Of relevance is McKinsey's 7S Model [34], comprising seven dimensions, to address the critical role of coordination in organizational effectiveness: structure, strategy, systems, skills, style, staff, and shared values. This has been used as the basis for the development of other models, such as Hanafizadeh and Ravasan's [35] readiness model for enterprise resource planning (ERP) systems. Cultural issues have also featured strongly in some studies of strategy alignment [6], and the balanced scorecard [36] has also been used in the context of IS strategy development. Balafif and Haryanti [37], for example, adapted the balanced scorecard dimensions to meet their research objectives and used the new defined IT balanced scorecard to measure success through defined KPIs. Bricknall et al. [38] similarly used the balanced scorecard to align IT strategy with business strategy in a multinational pharmaceutical company.

Building upon these concepts in the extant literature, the conceptual framework comprises two main dimensions: IS strategy change components and the IS strategy process phases. These are outlined and discussed below.

### 3.2. The IS Strategy Change Components

A distillation of the range of concepts in the extant literature suggests six key change components relating to IS strategy. These were initially formulated as the acronym COCPIT—Cost and Benefits; Organization and Processes; Human Capital; Systems Projects; Integration with Business Strategy; Technology Infrastructure [39]. However, on the basis of interview feedback, the labels were amended slightly and reordered to reflect a more logical chronological sequence, as detailed below.

Business strategy requirements are a key driver of IS strategy development and constitute the lynchpin upon which the on-going success of IS strategy implementation hinges. It is of particular significance in the alignment process phase (discussed below), providing the basis for the integration of business strategy and IS strategies. Coordinating and adjusting mechanisms may be introduced to encourage cooperation and resource sharing between subsidiaries and departments to engender alignment of IS and business strategies.

System projects are the main tool for strategy implementation and require appropriate planning, funding and resourcing, and management. Project management methodologies should be established and may include more formal approaches as well as agile methodologies, depending on the type of IS being implemented.

Technology infrastructure supports IS strategy implementation, providing the physical and regulatory framework for hardware, software, data management, and communications technologies. Standards may already be in place as part of the overall IT governance function. Similarly, standards and policies regarding system development and run-time environments—be they on premises or one of the variations of cloud environments—need clarity and appropriate documentation.

Process change encompasses organizational impacts and new ways of working associated with the new IS. This is part of what Galliers and Leidner [27] termed a "change management strategy", to manage process change as new systems are implemented. This component encompasses all company business processes designed to create value for customers (goods and services) and how they might change with the introduction of new IS.

Skills and competencies assess the likely changes in skills, culture, and human resources necessitated by the new IS. It considers cultural change issues, what characteristics and skills are needed when selecting people, how to develop global talent, how to hold meetings most effectively in a virtual environment, and how to manage employees and work with local or decentralized teams. This may be part of a corporate change management strategy, noted above, to provide support for new system users in a changed process environment.

Costs and benefits are central to IS strategy development and implementation. Costs need to be correctly identified and estimated. There are a variety of different cost elements, but capital project costs and on-going revenue expenditure are likely to feature in this analysis. Benefits are likely to be seen in terms of tangible revenue gains (increased market share, turnover, and efficiencies) and cost savings, as well as the intangible "soft" benefits, such as improved staff morale, cultural positives, and improved decision-making. This component is likely to be particularly evident in the control phase of the IS strategy process.

### 3.3. The IS Strategy Process Phases

The process phases build upon several of the existing frameworks discussed above. IS strategy development comprises two phases—review and align. Once they are completed, IS strategy implementation follows with the engage and execute phases. There follows an additional control phase to ensure quality and facilitate lessons learned before a completely new planning cycle starts again. The five phases are thus review, align, engage, execute,

and control, for which the acronym RAEEC is used. Depending on the size and structural complexity of the MNC, each phase will take varying time periods to complete, based on the company's planning cycles and the nature of the IS strategy being developed.

### 3.3.1. IS Strategy Development Phases

The review and align phases aim to ensure (continued) alignment between subsidiaries and headquarters. Each phase comprises multiple tasks that can be adjusted based on the company's global setup, and the political and cultural environments of the subsidiaries.

The review phase considers the corporate business strategy and a range of both internal and external factors. It provides a summary of the effectiveness of the present IS strategy based on goals or project KPIs and may also include a GAP analysis of current vs. planned project status. An analysis of the latest technology trends and their impact on existing strategies is also likely to feature in this phase.

The align phase evaluates the current state of information systems deployment in the company against current and future business requirements, based on the corporate strategy and subsidiary or departmental objectives. Cultural issues that might affect the future IS plan are identified. The IS strategy is adjusted based on strategic objectives and reworked through the six change components discussed above. Top-line success measures are defined or re-defined.

### 3.3.2. IS Strategy Implementation Phases

The engage phase determines the performance level of the organization as regards the current IS strategy, and identifies areas of concern and organizational barriers. This phase can be seen as a preparation for project initiation or re-launch and prepares the organization for the upcoming changes in systems, processes, and people skills.

The execute phase is at the core of the implementation process. An implementation roadmap can identify and manage integration issues and put in train system deployment and process and people change aspects. Appropriate project management methodologies are adhered to, using success criteria and KPIs as appropriate.

The control phase validates and verifies the implementation of the IS strategy based on the overall and project-specific success criteria set out in the Align and Execute phases. This can be done by internal or external audits or by an internal project management office function.

The conceptual framework is a combination of the change components and RAEEC phases. In a sense, the matrix represents the "what" (components) and the "how" (phases) of IS strategy development and implementation.

## 4. Results

### 4.1. Framework Review

Following the development of the conceptual framework, a questionnaire was designed to elicit information from interviewees that would provide a response to the research questions and allow the population of the conceptual framework to provide appropriate comments and actions to advance the strategy development and implementation cycle. The questionnaire contained 23 questions (Figure 3) and was emailed to the interviewees in advance of the interviews.

The data collected from the 18 in-depth expert interviews were analyzed, coded, and classified, as noted in Section 2 above. Similar responses were grouped together based on the IS framework components and phases. The responses to four questions that concerned respondents and company details were excluded. Other responses that were of a general nature or void were also excluded. This left 192 responses garnered from the questionnaires and the interview transcripts. These were assigned to the appropriate component/process cell in the conceptual framework.

| # | Questions |
|---|---|

**Respondent Details**

**Q1**     •   Can you outline your role in the organization please?
**Q2**     •   How many years' experience do you have in this role and in the company?
**Q3**     •   How many years do you have in a management position?
**Q4**     •   Have you ever been part of IS strategy development and/or implementation?

**The Business Planning Process and IS Strategy**

**Q5**     •   How is business strategy developed and implemented? (Especially regarding the role of headquarters and subsidiaries in this process)
**Q6**     •   How is IS strategy linked to the overall business strategy?
**Q7**     •   Which departments are involved at the headquarters and subsidiaries in both the development and the implementation of IS?

**IS Strategy Development**

**Q8**     •   How is IS strategy developed? (Especially regarding the role of headquarters and subsidiaries in this process)
**Q9**     •   Who is defining the IS strategy and who is leading the process?
**Q10**     •   How are cultural issues (e.g., know-how, skills) considered in IS strategy development?
**Q11**     •   How are internal and external factors considered in IS strategy development?
**Q12**     •   What is the IS strategy? What is the software policy for personal productivity tools (e.g., MSOffice) and for main business systems (e.g., SAP)?

**IS Strategy Implementation**

**Q14**     •   How is IS strategy implemented? (Especially regarding the role of headquarters and subsidiaries in this process)
**Q15**     •   Is a particular project management methodology used (e.g., PRINCE2 or PMI)?
**Q16**     •   Is there a clear business case for IS strategy implementation: Is there a cost-benefit analysis prior to implementing a particular software product?
**Q17**     •   Can you identify any key issues which are driven from IS strategy implementation?
**Q18**     •   What has been implemented in the past 5 years?
**Q19**     •   What is planned to be implemented in the next 5 years?

**IS Strategy Review**

**Q20**     •   How is the success of IS strategy development measured? Are the benefits clearly identified?
**Q21**     •   How is this strategy reviewed? Is there a process for amending the IS strategy if necessary?

**Varia**
**Q22**     •   Is there other information you can provide related to IS strategy development and implementation in your company?
**Q23**     •   Anything else you would like to add?

**Figure 3.** Questionnaire used in phase 2 of the research.

The allocation of the coded responses across the components and phases of the framework indicated some interesting variations in perceived emphasis and significance among the interviewees (Table 4). As regards the change components, the process change component received the most comments, followed by the systems projects component, indicating that the main thrust of an IS strategy is to put new systems in and change processes accordingly. In terms of the RAEEC phases, the align and execute phases received the most comment from the interviewees, which suggests the main concerns are to align strategy across the group and focus on progressing implementation via appropriate systems projects. The Align phase mainly drew comments as regards process change and technology infrastructure provision, emphasizing the need to get these aspects underway and planned for prior to strategy implementation. Other points of interest included the concentration on skills and competencies in the engage phase, indicating the need to upskill prior to embarking on new systems projects, and the distribution of process change feedback across all RAEEC phases. More specifically, perspectives coming from the interviewees (coded P1–P18 in Table 2) regarding the change components included the following points:

**Table 4.** Feedback from the 18 interviews in the RAEEC framework.

| PHASE ⟹ COMPONENTS ⇩ | Review | Align | Engage | Execute | Control | Total |
|---|---|---|---|---|---|---|
| Business Strategy Requirements | 3 | 5 | 4 | 4 | 1 | 17 |
| Systems Projects | 3 | 1 | 3 | 27 | 7 | 41 |
| Technology Infrastructure | 2 | 23 | 2 | 1 | 2 | 30 |
| Process Change | 12 | 30 | 10 | 10 | 4 | 66 |
| Skills and Competencies | 2 | 1 | 12 | 6 | 1 | 22 |
| Costs and Benefits | 1 | 4 | 1 | 4 | 6 | 16 |
| Total | 23 | 64 | 32 | 52 | 21 | 192 |

Business strategy requirements: Interview data were mainly coded to the align, engage, and execute phases (Table 4), which was to be expected as this component focuses on business alignment and corporate governance, which are of importance in ensuring a successful and effective IS strategy implementation. Feedback also noted that the local political system and government policies will have an impact on the nature of IS strategy implementation in subsidiaries. P3, for example, noted, "it is essential for us to monitor upcoming trends and megatrends to analyze the political impact they might have at subsidary level". Local government, reflecting its political orientation, may take a hands-off approach towards IS developments. On the other hand, despite all good intentions, the government may impose a wide array of overly restrictive policies. Equally, the government may pursue an aggressive policy of rapid technology growth and provide necessary incentives and infrastructure to enterprises for technology investment. The adoption of cloud services was cited as a useful illustration of the need for a control phase to contain and manage the proliferation of cloud-based IS activities in an enterprise. The control phase may entail compliance audits, which can be internal or external, although, according to P17 and P18, external auditors will better guarantee an independent perspective. All interviewees maintained that a good understanding of the global IS environment will be a crucial factor in the development of a suitable IS strategy for a global enterprise. A central top-down approach can achieve the best results and avoid the development of independent software choices and operations, with consequent problems of connectivity, integration, and data consolidation. P12, for example, reported that "we use a global template approach for our ERP systems which is stricly driven by a top-down approach". In similar vein, P14 stated that "the core components of our ERP applications are configured top-down. For our treasury business, it causes some challenges as we also need to be compliant with local financial policies".

Systems projects: Based on the coded distributions across the RAAEC phases, the focus is clearly on the execution of the IS strategy, with 27 comments (Table 4). Interviewees highlighted the need for an appropriate methodology for IS implementation, with a recognized project methodology in a multi-national context being the most frequent code identified in the Execute phase. On the other hand, there was just one code that linked to the align phase, which made reference to the choice between long-range strategic considerations versus shorter-term agile strategies, observing that "it is essential for my department to have a long-term strategic vision, but as technology and new applicartions are rapidly changing, it is very difficult for me to find the right balance. Aligning the IS strategy with current market trends is also essential". This decision point has significant implications for the methodologies chosen, the required skills for project team members, and the nature of the change management process. The need to address local country issues and culture was also highlighted, notably by P1 and P15. Elements that can influence project outcomes in the subsidiaries include economic growth, national culture, and the political system as causal factors, among others. If culture is identified as one of the factors influencing IS needs, then it can be explored in more detail, both in terms of specific cultural features and the requirements for IS customization. In this context, P17 opined that "based on my experience, culture is nearly always different in every country, and our managers have to keep this in mind. Project execution can be delayed because of different understandings and norms".

Technology infrastructure: Issues concerning technology infrastructure were concentrated in the align phase as the implications of system rollout plans were considered and the need for technology infrastructure upgrades was assessed. Responses concerned three main technology topics: the adoption of cloud technologies and implications for IS strategy; software choices and trends; and data management issues. In all six companies, cloud services were considered a central part of the IS strategy. They all differ somewhat from each other in their use of the cloud, and the services used reflect the complexity of their current IT architectures. Due to company size and the small number of subsidiaries, companies B and D (Table 1) were able to adopt a cloud-only approach for their applications. The other companies are using a hybrid or a private cloud approach because of the complexity of their organizations. In terms of overall technology integration and connectivity, Company B uses a multi-domestic strategy, meaning each subsidiary can define its own strategy and policies to some extent. This provides the subsidiaries with local flexibility and agility but complicates the implementation of other elements of the global IS strategy and means different cloud deployment models are utilized. A main challenge in deploying cloud-based solutions involving company data is trust, and this challenge is heightened in an MNC using multiple cloud environments. Data are an organization's most important asset, and delegating the management of that data to a number of different cloud suppliers requires confidence in the providers' technology solutions, underlying infrastructure, expertise, security, and credibility in delivering bulletproof mission-critical solutions in what are often highly regulated industries.

Process change: Effective process management can provide the necessary flexibility to adapt an organisztion to new ways of working as an IS strategy is implemented. If appropriate, standard processes can be adopted throughout the subsidiaries of the organization and be aligned with the overall strategic intent of the enterprise. In this context, P5 summarized the position in company A: "By linking process know-how to the roles of individuals and incorporating it into competency metrics, we could create a framework that not only facilitates training but also clarifies the distinctions between roles. This approach can streamline operations and ensure that everyone understands their responsibilities and how they contribute to the overall goals of the company". Interviewees indicated that process change needs reviewing across the RAEEC phases, but notably in the align phase, as preparations are made for system implementation. Multi-national enterprises can be classified based on whether the end users of their products are industrial customers, individual consumers, or internal holding companies, and each of them requires different

processes that affect the IS, and vice versa. Nevertheless, common to most companies' operations are two sets of business processes, front- and back-office processes, which can be very complex due to the structure of the enterprise and the degree of centralization (vs. decentralization) of the organization and its strategy. Back-office processes are more amenable to global coordination because unique front-office processes are required to tailor products for different markets. Enterprises generally make larger capital investments related to back-office processes compared to front-office processes, with the objective of reducing the long-term cost of back-office operations.

Skills and competencies: Issues raised by the interviewees regarding skills and competencies related in the main to the execute and, above all, the engage phases reflect the need to review skills requirements and plan and undertake training and/or recruitment prior to system implementation. The rapid advancement of systems technologies requires resources to develop, maintain, and sustain capabilities, and this is likely to be challenging for smaller local subsidiaries. The IS strategy in an MNC must consider the availability of people, skills, and know-how while at the same time being aware of cultural influences. P9 explained the significance of the alignment of competencies with IS strategy: "With a strong emphasis on core competencies and aligning them with the strategy, especially in the context of Industry 4.0 and the evolving landscape of technology, we were positioning our company to stay competitive and relevant. Addressing aging IT infrastructure and leveraging emerging technologies will be crucial in achieving your objectives".

P1 noted that part of his role as CIO was to act as a mentor to engender cross-cultural teamwork and engage team members from many regional subsidiaries to work together. Systemic resistance results from the passive incompetence of the enterprise in support of the strategy, and arises whenever the development of capacity lags behind strategy implementation. The adaptability of organizational culture to accommodate strategic change is an essential measure in overcoming this challenge. IT manager P13 also emphasized the importance of transparency at the board level, noting that it is "crucial to establish protocols for escalation and decision-making in exceptional cases where certain parties may question the board's decisions or competencies. This ensures transparency, accountability, and effective decision-making processes within the organization". The key is to maximize synergy while reformulating a new organizational culture to reflect the wider change brought in by new systems and associated processes.

Costs and benefits: Interviewees made some reference to cost and benefit issues across all RAEEC phases, but notably in the control phase. Operations manager P18 noted that "for larger projects, there's a structured approach in place where thorough analyses are conducted, including cost-benefit analyses, before implementation. This ensures that decisions are made based on a comprehensive understanding of the potential benefits and impacts on the organization". There were two main aspects here: the more formal commitment to tangible and intangible project benefits, involving revenue and capital costs, and the assessment of cross-company value chain activities. Defined business cases can be used to measure costs and benefits, but a challenge in some MNCs is the lack of transparency in some matrix organizations within group companies. Value chain coordination, on the other hand, refers to deploying IS to coordinate similar activities (such as procurement or production) across different geographic locations through centralized processes that increase efficiency and bring flexibility. This allows the improved availability of information through various transmission channels to be transfered and absorbed by headquarters and subsidiaries at lower costs. Challenges due to different working behaviors and business models within holding companies should be addressed accordingly. By optimizing its value chain activities, an enterprise can achieve efficiencies through centralized administrative coordination, control of resources, and performance measurement. The success of value chain activities can be measured by business process efficiency increases.

*4.2. Case Study Development of an Action List for the RAEEC Framework*

The framework developed from the literature review was applied and tested in a mid-size multi-national group company selected because the research team had well-developed contacts within this organization. The company's core business is air quality technology, creating awareness as well as providing practical solutions that help create living and working environments that are safe, healthy, and enjoyable.

The company was founded in 1963 in Germany, but in the year 2000, the company moved from Germany to Switzerland, from where the company has operated since then, with research and development, production, sales, and product management functions, and a global headcount in 2023 of 500 employees. In 2005, a US subsidiary was founded to function as a sales entity for the North American and Canadian markets. A further subsidiary was set up back in Germany in 2014 as an additional production plant, and another retail subsidiary was set up in China in 2015 for the Chinese market. The Swiss company retains responsibility for other world markets. The company was expanding in Germany and China at the same time, and there was no overall IT or IS strategy. Each entity made independent decisions regarding IS strategy, and the group's setup engendered a decentralized approach, with each company having a different ERP system and using cloud services from different suppliers. In 2016, the CEO and the senior management concluded that a more coordinated approach to IS strategy would be beneficial, and initiatives were taken to transition to a centralized approach, driven by the Swiss headquarters.

The RAEEC framework was used to develop and implement a new IS strategy for the group in the period 2020–2022. This was a two year cycle that involved the selection and implementation of a new ERP system, which meant that the engage and execute phases, in particular, took an extended period of time. The research team presented the framework to the senior management team in 2020, when a detailed introduction was provided. The review phase started in Q2 2020 and took approximately three months, resulting in a joint agreement across the companies to find and implement new integrated systems for the group—a new ERP system. The align phase started in Q3 2020 and involved the detailed development and definition of a new IS strategy. This overlapped with the start of the engagement process, which began in Q1 2021. The execution phase was the most time-consuming and challenging. The implementation of the new ERP system was completed in the United States and China subsidiaries by the end of 2021, and in Q1 2022, the old ERP systems were decommissioned. In the Swiss and German companies, data analysis and migration preparation activities throughout 2022 delayed going live with the new ERP system until Q1 2023. The control phase took place across the different companies, spanning 2022 and 2023, reflecting the varying implementation timescales.

The overall process was observed and recorded by the research team as it unfolded, and detailed notes of actions and decision points were taken. From this, a set of generic actions for MNCs in their use of the RAEEC framework was developed (Table 5). This incorporated indicators that were evident in the feedback from the 18 interviewees and also built upon discussion points in the existing literature. The action list is thus built from three main sources—the interviews, the case study application, and the existing literature. This is not seen as a definitive list but rather as representing a starting point for a management team embarking on IS strategy development and implementation in an MNC environment.

**Table 5.** Action list for RAEEC framework.

| Component/Phase | Review | Align | Engage | Execute | Control |
|---|---|---|---|---|---|
| Business Strategy Requirements | • Review current business strategy and any updates.<br>• Examine IS strategy against updated corporate business strategy. Identify areas of change and investigate where mismatches may be in evidence in subsidiaries.<br>• Top line review of new business requirements in subsidiaries.<br>• Assess local political issues and their potential impact in subsidiaries. | • Update and amend specification of business requirements in subsidiaries.<br>• Match new and ongoing systems projects to evolving corporate business strategy.<br>• Consider and accommodate variations in IS technical environments across subsidiaries.<br>• Adjust strategic dimensions of IS strategy, Undertake software selection process as appropriate.<br>• Define/redefine success measurements. | • Review relevant corporate governance issues (e.g., cybersecurity and data protection).<br>• Assess national/local government regulation and subsidy opportunities from local agencies and government bodies.<br>• Inform board members about upcoming projects and changes. | • Adopt a top-down approach to IS strategy implementation, but not to the total exclusion of bottom-up initiatives.<br>• Deliver and communicate high-level tasks and implementation milestones (Executive/Board level). | • Validate and analyse IS Strategy success criteria measurements.<br>• Undertake audits (internal and/or external) of critical elements of systems strategy in line with corporate governance requirements. |
| Systems Projects | • Audit all ongoing systems projects against project briefs/plans—timescales, budgets, cost and benefits etc.<br>• Assess new or recent systems technology trends and the impact of digital products deployment. | • Revise systems project plan across organisation to reflect new or amended IS strategy.<br>• Consider "local" issues—the national context where subsidiaries operate; culture issues.<br>• Define the use of working tools and methodologies for systems projects.<br>• Set technology standards, procedures, and guidelines. | • Assemble/reassemble project team(s). Agree methodologies and working practices. Create project goals and regular effectiveness reviews.<br>• Create measurements and key indicators for project teams.<br>• Establish communication channels within teams and with technology support.<br>• Highlight the importance and the effectiveness of the project structure.<br>• Evaluate training needs and plan training/skills programs (for both IT staff and end-users).<br>• Develop implementation roadmap at subsidiary/project level. | • Configure and apply mainstream project management methodology, but allow for bottom up initiatives that may use agile methods.<br>• Implement software and support infrastructure, working with technical support teams.<br>• Consider and accommodate (as appropriate) change requests driven by local requirements at subsidiary level. | • Assess project progress against KPIs, milestones, benefits delivery.<br>• Revisit upward and downward reporting; escalate key issues.<br>• Develop team dynamics and project ownership.<br>• Verify and celebrate implementation successes. |

**Table 5.** *Cont.*

| Component/Phase | Review | Align | Engage | Execute | Control |
|---|---|---|---|---|---|
| Technology Infrastructure | • Evaluate current state of technology infrastructure across subsidiaries.<br>• Review new infrastructure and cybersecurity technologies, and procurement options.<br>• Assess any change in technology standards that may impact systems projects. | • Assess technology infrastructure requirements to support ongoing IS rollout to include networks, servers, cloud services.<br>• Review cross-company governance and compliance issues for cybersecurity, data privacy and protection, and AI deployment. | • In conjunction with technology managers in HQ and in subsidiaries, develop/update fully costed technology infrastructure plan.<br>• Prepare systems and data migration plans. | • Implement detailed infrastructure plans, based on systems milestones.<br>• Upgrade network and storage capacities as required.<br>• Adjust backup and business continuity plans. | • Monitor network performance.<br>• Monitor cloud services operational issues across subsidiaries. |
| Process Change | • Review and update cross-company process maps and process definitions.<br>• Assess process ownership at central and local levels.<br>• Identify areas of concern (process bottlenecks, tensions, inefficiencies). | • Define processes using systems templates as appropriate.<br>• Map processes to systems modules/systems projects.<br>• Document processes in need of change or improvement. | • Assess process efficiency at subsidiary level.<br>• Identify organizational barriers to process improvement.<br>• Plan for process change implications. Staff roles, working practices, training.<br>• Put in place IS support process (local and centralized). | • Reinforce user process ownership.<br>• Introduce process change in conjunction with systems deployment.<br>• Support and provide training for affected individuals.<br>• Document new processes. | • Monitor process change.<br>• Sign off process documentation.<br>• Drive through process change benefits.<br>• Execute re-audit process to pass required certification (e.g., ISO 2700x). |
| Skills & Competencies | • Review project team performance across subsidiaries.<br>• Review the skills requirements of project team members and central IT function.<br>• Review role descriptions in IT and in user functions involved in systems projects.<br>• Identify cultural or language barriers evident in systems projects. | • Cascade people related principles throughout organisation and highlight the importance of teamwork.<br>• Clarify company guidelines and policies on HR issues.<br>• Identify and support language weaknesses.<br>• Create cross-cultural/cross subsidiary workshops. | • Undertake performance reviews with all staff to set agreed and coordinated objectives.<br>• Identify and address cultural issues and resistance to change.<br>• Undertake/review appraisal/staff development activity.<br>• Organize cultural training—and use as an ice breaker.<br>• Define support actions and the steps to be taken if escalation required.<br>• Identify training needs.<br>• Plan for IT support staff recruitment and skills enhancement. | • Initiate change management process.<br>• Create an environment where teams can focus on systems delivery.<br>• Manage expectations of digital natives. | • Monitor staff performance.<br>• Recruit replacement staff as necessary.<br>• Ensure team members leave the systems projects feeling appreciated.<br>• Consider opportunities for staff rotation between subsidiary projects.<br>• Check planned training completed successfully. |

**Table 5.** *Cont.*

| Component/Phase | Review | Align | Engage | Execute | Control |
|---|---|---|---|---|---|
| Costs & Benefits | • Undertake annual (and periodic) review of overall capital and revenue costs on systems projects, at subsidiary and company level.<br>• Quantifiable benefits assessment as systems implemented to be fed into final cost-benefit analysis. | • Cross-subsidiary allocation of costs and benefits, notably for large systems modules.<br>• Allocate budgets for required capital expenditure items and revenue costs.<br>• Progress centralised financial consolidation of systems projects. | • Detailed planning of systems projects costings (software, other technologies, staffing and revenue costs). | • Manage financial aspects of technology acquisition (software and all associated infrastructure).<br>• Ensure cost transparency.<br>• Use new systems to adjust cost structures and reporting as appropriate. | • Control costs via monthly/periodic expenditure monitoring.<br>• Monitor and record benefits delivery against project targets (increased sales, reduced costs, etc.). |

*4.3. Discussion*

The interview findings and case study application highlight a number of issues relating to IS strategy in MNCs that are worthy of further discussion. First, the findings underscore the importance of alignment in IS strategy development, particularly for technology infrastructure and business processes. This resonates with recent research by Ilmudeen et al. [40], who concluded that "failure to maintain alignment between business and IT strategies hinders performance outcomes", and that "this occurs when a firm fails to properly manage and govern IT investment" (p. 214). The RAEEC framework can be used as a practical vehicle for promoting alignment of IS and business strategies (notably via the align phase), and for ensuring IS budgets are appropriately managed to support IS initiatives and projects (via the costs and benefits change component and control phase).

Second, the findings suggest that digitalization can be encompassed within IS (and IT) strategy development and implementation and does not require separate treatment as part of a "digital transformation strategy" [41] or "digital transformation roadmap" [42]. Stock and Teubner [43] (p. 4) noted that "recent studies indicate that coping with the challenges induced by digitalization is right at the top of IT/IS strategy agendas in practice and that IT executives actively seek advice on managing digitalization strategy". The findings suggest digital transformation can be incorporated into a standard IS/IT development and implementation framework, such as the one put forward here. This accords with the view that digitalization has to date largely resulted in evolutionary rather than revoltionary change [44], and associated innovation is more often incremental than disruptive and only on occasion results in a significant change of business model.

Third, and related to the above, the suggestion put forward by some researchers [45,46] that IT/IS strategy should be merged with an overall digital business strategy is not supported by the feedback from practitioners in the MNC companies studied here. While such an approach may be appropriate for smaller companies, particularly those either developing or sourcing digital products or services, it is likely to be unworkable in the complex business and technology environments often encountered in MNCs. Teubner and Stockinger [47] (p. 33) observe that "a comprehensive and coherent IT/IS strategy, rather than being rendered obsolete, may become more vital than ever in an increasingly digitalized world", although "this is not to deny the close and ever-increasing intertwinement of IT/IS and business strategy in digital business".

Fourth, the findings support the results of other research on IS and IT strategies in multi-national industries that suggest certain key competencies emerge as being of prime importance for successful IT and IS implementation in the digital age. For example, Wynn and Lam [48], in their study of four major international hotel groups, found that process agility, workforce adaptability, and technology mangeability stood out as key requirements for successful IT/IS implementation. In the context of this research, process agility involves not only re-engineering processes to take advantage of system benefits but also remaining open to continuous reinvention and change in working practices, notably in the align, engage, and execute phases. Workforce adaptability—sometimes termed workforce orchestration [49]—encompasses not only the re-skilling and recruitment of staff as necessary as new systems projects are planned and implemented in the engage and execute phases, but also the mindset to continually improve and adopt new skills and adapt to new cultural norms or attitudes. Technology manageability is a key condition for successful IS implementation, and the scope and integration of systems projects need careful planning and assessment in the review and alignment phases. The provision of customer-centered system innovation must be founded upon a stable and manageable technology platform that can provide consistent data to support effective cross-company decision-making and contain cybersecurity management issues. This is often particularly challenging for MNCs that may have acquired or merged with other enterprises in which legacy systems and outdated technology infrastructures remain.

## 5. Conclusions

This article used an analysis of the pertinent literature, a set of questionnaire-based interviews, and an applied case study to address three research questions relating to the IS strategy in MNCs. A short summary response to each question is presented here in three sub-sections, followed by a brief discussion of the limitations of the research and possible avenues for future research projects in this field.

### 5.1. From an Analysis of the Extant Literature, What Conceptual Framework Can Be Constructed to Guide IS Strategy Development and Implementation in MNCs? (RQ1)

The extant literature contains a range of concepts and themes relating to IS strategy, including those put forward by leading academics and practitioners over recent decades [9,22,23,25]. This research attempted to take a holistic view of IS strategy development and implementation in MNCs, distill the main concepts of relevance, and work them up into a conceptual framework that could provide the basis for an operational model that would be of relevance and value in practice. This produced the two-dimensional conceptual framework discussed above, comprising six change components and five process phases.

The development of this framework was an iterative process spanning several months in which the definition and labeling of the concepts were refined and adjusted as the literature was cross-checked and the framework was discussed and evaluated within the research team. The framework (the "RAEEC framework") is depicted in Table 4, supported by the action list contained in Table 5.

### 5.2. What Does Feedback from Practitioner Interviews Indicate Regarding the Relevance and Usability of the Framework? (RQ2)

The RAEEC framework was used to analyze feedback from 18 industry practitioners working in MNCs, who answered a range of questions regarding the development and implementation of IS strategies in their organizations. Overall, this suggested that the framework is a valid mechanism for classifying and analyzing IS strategy components and processes and that it could usefully be employed as a top-line operational model for IS strategy development and implementation in practice. Feedback was classified and allocated to the appropriate cells in the framework matrix (Table 4). This indicated a reasonably equitable distribution across change components and process phases, but certain pinch points were identified where additional focus and resource allocation may be required. Of particular significance were the need to plan for technology infrastructure and process change in the align phase of strategy development and the focus on systems projects in the execute phase. What is also relevant is the need to adjust skills and competencies in the engage phase prior to project implementation and the significance of monitoring and managing the process change requirements right across the RAEEC phases.

These findings are of no great surprise, but nevertheless suggest the RAEEC framework is relevant to the challenge of developing and implementing IS strategies in the MNC environment and may thus be of value to practitioners confronting the wide range of associated issues in a rapidly evolving technology environment. There is no set timeframe for progressing through the five phases of the RAEEC framework. In the case study company, this spanned a two year period—to go through the full five-phase cycle—but this entailed the replacement of the core ERP system, which would normally occur every 10 years or so, or longer in some cases. Once the core elements of an IS strategy are in place (such as the ERP system, the main network infrastructure, the support personnel, and processes), this cycle will more likely be undertaken and completed on an annual or biannual basis and be part of a coordinated, wider business strategic planning process. Another aspect here is that the implementation phases (engage–execute–control) will be more or less continuous, and individual projects and different subsidiaries will be at different stages in the overall cycle. The framework can nevertheless be used flexibly to guide IS strategy implementation across an MNC and its subsidiaries.

*5.3. Can a Set of Key Actions be Developed to Support the Use of the Framework in Practice? (RQ3)*

Based on the feedback from the 18 interviews, plus the application of the framework in the case study company, a set of actions was developed to support the application of the framework and its use as an operational model (Table 5). Of necessity, these are generic in nature, as they are intended to be relevant to the diverse environments encountered in MNCs. The set of actions is best seen as a starting point to initiate the IS strategy development and implementation process. They are possible actions, not mandatory, from which the most appropriate can be selected, customized and applied in different MNC business contexts.

*5.4. Limitations and Future Research*

This research has its limitations. The framework is a logical development from the extant literature that was applied retrospectively to the interview material gleaned from 18 industry practitioners. This was a form of validation, but only in one case study was the framework applied. Nevertheless, it constitutes a top-line working model that can be applied flexibly, along with the action list, in different MNC environments. As such, the authors believe it is a valid contribution to the literature that can be set alongside other models relating to alignment and change in MNCs.

Future research could test out and develop the framework in other MNCs, possibly in combination with other methodologies already in use that can be accommodated within the umbrella RAEEC framework. Previous studies have examined how leading project management methodologies can be adapted to the SME environment [50], and this could be extended to examine how they can best be integrated within the RAEEC framework for use in the MNC environment. The framework could also be adapted and extended to encompass all IT, rather than having an information systems focus, and thus take on board the multiple implications of digitalization, including broader cybersecurity and governance issues, which pose particular challenges for MNCs operating in a number of different countries. The action list could also be further developed to include more specific actions of relevance in different MNC environments, emphasizing that this is not a definitive mandatory list, but rather a set of options which can be applied selectively depending on the particular MNC business and technology environment. Stockhinger and Teubner [43] (p. 4) noted that "we know little about strategy contents in the digital age so far", and that "as long as actual topics, issues and concerns of strategic IT/IS planning are missing, planning methodologies and concepts necessarily remain vague and practical implications remain limited". It is hoped that this article, in providing a framework based on practitioner perspectives, has made a small contribution to addressing this gap in the research literature.

**Author Contributions:** Conceptualization, M.W. and C.W.; methodology, M.W. and C.W.; software, C.W.; validation, C.W.; formal analysis, M.W and C.W.; investigation, C.W.; data curation, C.W.; writing—original draft preparation, M.W. and C.W.; writing—review and editing, M.W.; visualization, M.W. and C.W.; supervision, M.W. and C.W.; project administration, M.W. and C.W. All authors have read and agreed to the published version of the manuscript.

**Funding:** This research received no external funding.

**Institutional Review Board Statement:** Not applicable. No humans or animals were studied, and thus, the university Ethics Committee did not need to be consulted.

**Informed Consent Statement:** All interviewees agreed to the use of their anonymized responses and feedback for research purposes.

**Data Availability Statement:** The original contributions presented in the study are included in the article; further inquiries can be directed to the corresponding author.

**Conflicts of Interest:** The authors declare no conflicts of interest.

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
