# Peer review of "Information Systems Strategy for Multi-National Corporations: Towards an Operational Model and Action List"

_information, doi:10.3390/info15020119_

Round 1

Reviewer 1 Report

Comments and Suggestions for Authors

This article proposes a model to develop and implement an information systems strategy in multi-national corporations (MNC). For that purpose, the authors rely on a qualitative approach by performing a literature review and interviews with 18 industry practitioners employed in six MNC and a case study to define an action list. This procedure identifies six change components and five phases. The authors identify three core dimensions of an IS: organization, management, and technology. Furthermore, the authors analyze the IS strategy change components and the IS strategy process and propose a list to assist the implementation of the model.

The structure of the article is correctly chosen and it exhibits a logical concern. The methodology is well explained and implemented. The text is written in perfect English, but, some typos have to be removed, such as in line 373 remove the second is, in line 420 correct availability, line 440 correct was, and in line 489 remove bold from and.

The authors come up with interesting and useful insights to define the Information Systems strategy for MNC. However, I noticed that most of the literature reviewed is based on old articles, with only 6 relating to the last 5 years. Then I would recommend some literature updates.

Reviewer 2 Report

Comments and Suggestions for Authors

Dear/s Author/s,

Re: Manuscript “Information Systems Strategy for Multi-National Corporations: Towards an Operational Model and Action List

Reviewer’s report:

The topic is interesting as it is about enriching the information systems strategies of multinational corporations at a time when most of the decision making is done through these systems. The research gap is well highlighted. The paper is well written and structured. Within the databases, neither Scopus nor Web of Science, which are the most relevant at world scientific level, are observed, which should be exposed at the end as a limitation.

With respect to the number of interviews, I consider that it would be more correct to state that no more interviews were made, because a number was reached in which new interviews were not going to provide more information. It is necessary to provide more data on the coding criteria and to state them clearly in the methodology.

The interviews seem to be conducted on a global scale and the paper itself talks about cultural differences, but does not specify the culture to which the interviewees belong. If the interviews were conducted at a more local level, the case would have to be justified.

There is no discussion section that clearly shows the theoretical and practical contributions of the work. The conclusions are limited to answering the questions, and this is correct, but the contribution of the work could be clearer for the readers if there were a discussion section with the exposition of these contributions. These aspects need to be systematized to gain clarity.

The literature needs to be updated, since for such a topical subject, there are no articles from the year 2022 and 2023.

Best Regards

Author Response

Pls see uploaded file.

Reviewer 3 Report

Comments and Suggestions for Authors

Comments on the Quality of English Language

Moderate editing of English language required

Author Response

Pls see uploaded file.

Round 2

Reviewer 2 Report

Comments and Suggestions for Authors

Dear/s Author/s,

Re: Manuscript “Information Systems Strategy for Multi-National Corporations: Towards an Operational Model and Action List”

Reviewer’s report:

The authors have satisfied with their contributions the recommendations made.

Best Regards

Reviewer 3 Report

Comments and Suggestions for Authors

The authors have already revised the manuscript according to the reviewers’ comments.